# Effects of Intrabony Length and Cortical Bone Density on the Primary Stability of Orthodontic Miniscrews

**DOI:** 10.3390/ma13245615

**Published:** 2020-12-09

**Authors:** Jie Jin, Gi-Tae Kim, Jae-Sung Kwon, Sung-Hwan Choi

**Affiliations:** 1Department of Orthodontics, Institute of Craniofacial Deformity, Yonsei University College of Dentistry, Seoul 03722, Korea; kimj0515@yuhs.ac; 2Department and Research Institute of Dental Biomaterials and Bioengineering, Yonsei University College of Dentistry, Seoul 03722, Korea; gitae7673@yuhs.ac (G.-T.K.); Jkwon@yuhs.ac (J.-S.K.); 3BK21 FOUR Project, Yonsei University College of Dentistry, Seoul 03722, Korea

**Keywords:** miniscrews, miniscrew length, primary stability, intrabony length, removal angular momentum, cortical bone density

## Abstract

Miniscrews have gained recent popularity as temporary anchorage devices in orthodontic treatments, where failure due to sinus perforations or damage to the neighboring roots have increased. Issues regarding miniscrews in insufficient interradicular space must also be resolved. This study aimed to evaluate the primary stability of miniscrews shorter than 6 mm and their feasibility in artificial bone with densities of 30, 40, and 50 pounds per cubic foot (pcf). The primary stability was evaluated by adjusting the intrabony miniscrew length, based on several physical properties: maximum insertion torque (MIT), maximum removal torque (MRT), removal angular momentum (RAM), horizontal resistance, and micromotion. The MIT and micromotion results demonstrated that the intrabony length of a miniscrew significantly affected its stability in low-density cortical bone, unlike cases with a higher cortical bone density (*p* < 0.05). The horizontal resistance, MRT, and RAM were affected by the intrabony length, regardless of the bone density (*p* < 0.05). Thus, the primary stability of miniscrews was affected by both the cortical bone density and intrabony length. The effect of the intrabony length was more significant in low-density cortical bone, where the implantation depth increased as more energy was required to remove the miniscrew. This facilitated higher resistance and a lower risk of falling out.

## 1. Introduction

In recent years, miniscrews have been widely adopted by orthodontists as temporary anchorage devices due to their small size and minimal surgical intervention. Miniscrews can be implanted in different areas of the alveolar bone, which provides a broad range of options in orthodontic treatments. Furthermore, miniscrews have lower requirements for patient compliance and lead to less discomfort than traditional anchorage devices [1]. However, the most recent meta-analyses indicate that the miniscrews used in clinical treatment have a failure rate of 13.5 to 14% [2,3,4]. Primary stability is one of the critical properties affecting the success rate of miniscrews and is related to several factors, including the geometric design of miniscrews, condition of the host, operation techniques, and inflammation of the surrounding tissues [5]. A larger bone–implant interface area can be obtained by increasing the length and diameter of the miniscrew, thereby improving the primary stability. However, the pressure caused by high torque on the adjoining tissues can lead to necrosis of the surrounding bone [6,7]. Furthermore, long miniscrews may cause bicortical sinus perforations or damage to the neighboring roots [8]. Several host factors may also affect treatment success, where insufficient interradicular space is one of the main difficulties in miniscrew implantation. Specifically, inadequate support from the bone surrounding the miniscrews can compromise the stability of the miniscrew. Moreover, movement of the miniscrew during the treatment and contact with the roots can lead to miniscrew failure or even root damage [9]. The latter may be avoided by reducing the size of the miniscrew, but sufficient primary stability must be maintained. Different areas of the alveolar bone provide different spaces for the implantation of miniscrews, and a shorter miniscrew can provide more treatment options. Previous studies [10,11] have reported that there is no direct correlation between the length of a miniscrew and the success rate, but compromised mechanical performance may pose a risk [8]. However, most previous studies have employed miniscrews larger than 6 mm. Therefore, an in vitro study is necessary to evaluate the effect of the length of miniscrews when the insertion depth is less than 6 mm, which can then provide a basis for subsequent clinical studies. Polyurethane foam is one of the substrates often used to simulate human cortical bone and cancellous bone [12,13,14]. In biomechanical experiments, polyurethane foam can maintain homogeneity and consistency, which assists with the elimination of bias caused by other factors and exhibits stress–strain behavior similar to that of human bone, making it suitable for preclinical evaluation of miniscrews [12].

This study aimed to evaluate the primary stability of miniscrews shorter than 6 mm (3, 4, 5, and 6 mm) by changing the intrabony miniscrew length. The feasibility of the short miniscrews was assessed in artificial bones of different densities, which represented human alveolar bone. The null hypothesis was that there is no difference in primary stability between miniscrews shorter than 6 mm at different cortical bone densities.

## 2. Materials and Methods

### 2.1. Materials

Cylindrical-type orthodontic titanium miniscrews were used with no surface treatment (machined surface, OSSH1810, Osstem Implant, Busan, Korea) and a diameter, pitch, length, and head height of 1.8, 0.7, 10, and 1.95 mm, respectively (Figure 1A). According to standard procedure (ASTM F-1839-08), polyurethane foam blocks (Sawbone, Division of Pacific Research Laboratories, Inc., Vashon, WA, USA) with densities of 20, 30, 40, and 50 pounds per cubic foot (pcf)—equivalent to densities ranging from 0.32–0.80 g/cc—were used to represent cancellous bone and cortical bone (Table 1). This range of densities correlates with the human maxillary cortical bone density (ranging from 0.31–1.11 g/cc) reported by Devlin et al. [15] and the human mandibular bone density (average 0.664 g/cc) reported by Kido et al. [16]. The porosity of the polyurethane foam blocks was measured via micro-computed tomography (CT) (SkyScan 1076, Kontich, Belgium) (Figure 2). According to previous studies [14,17,18], the artificial bone comprised a 2 mm layer of higher density (30, 40, and 50 pcf) polyurethane foam representing cortical bone, and a 20 mm layer of lower density (20 pcf) polyurethane foam representing cancellous bone (Figure 1B). Miniscrews were implanted into the artificial bone at different depths (3, 4, 5, and 6 mm) (Figure 1C) using an automatic torque device (Admet eXpert8600, ADMET, Norwood, MA, USA) to control the insertion depth [19].

### 2.2. Torque

Torque testing was conducted according to a previously reported procedure [18,20]. The artificial bone was fixed with a metal clamp, and the miniscrews (*n* = 6) were implanted in a clockwise manner using an automatic torque device (Admet eXpert8600, ADMET, Norwood, MA, USA) with a self-drilling system at 5 rpm according to ASTM F-1839-08. The implantation depth (3, 4, 5, and 6 cm) was controlled by a supporting software (GaugeSafe software, ADMET, Norwood, MA, USA). Removal was performed in a counterclockwise manner at 5 rpm. The torque was measured every second during implantation and removal using the supporting software, and the maximum insertion torque (MIT) and maximum removal torque (MRT) were recorded.

### 2.3. Removal Angular Momentum 

The removal angular momentum (RAM) was evaluated to determine the energy required to remove the miniscrews [21] (*n* = 6). Angular momentum (Ncm·s) was calculated by integrating the torque over the first 12 s (one rotation) of removal.

### 2.4. Horizontal Resistance

Horizontal resistance was conducted according to a previously reported procedure [14,18]. A universal testing machine (Instron 5942, Instron, Norwood, MA, USA) was used to measure the horizontal resistance, where the artificial bone was fixed with a metal clamp while a knife-like shear jig applied a tangential load to the miniscrew (*n* = 6) at a crosshead speed of 1 mm/min (Figure 3). In the current measurement, the loading position was selected to be the same position, close to the upper part of the bone–miniscrew interface to minimize the deviation caused by the lever effect arising from the variation in the length of the free part. The displacement of the miniscrew and the applied force values were determined using supporting software (Bluehill 2, Instron Corporation; Norwood, MA, USA). The force at 0.6 mm displacement was recorded.

### 2.5. Micromotion

Micromotion analysis was conducted according to a previously reported procedure [18]. The periotest value (PTV) tester (Medizintechnik Gulden, Modautal, Germany) and implant stability tester (IST) (AnyCheck, Neo Biotech, Seoul, Korea) are two commercial hand-held micromotion measurement devices that obtain micromotion data by repeatedly tapping the head of a miniscrew. The artificial bone was fixed with a metal clamp, and the devices were set at a fixed distance from the miniscrew (*n* = 6) surface, in a direction perpendicular to its long axis. In the current measurement, the tapping position was selected to be the same position close to the upper part of the bone–miniscrew interface to minimize the deviation caused by the lever effect arising from the variation in the length of the free part. The devices were calibrated before each measurement according to the manufacturer’s instructions [22]. Measurements were performed in triplicate, and the mean values have been reported.

### 2.6. Statistical Analysis

All statistical analyses were conducted using IBM SPSS software (Version 25.0, IBM, Incheon, Korea). One-way analysis of variance (ANOVA) with Tukey’s test was conducted, where a *p*-value of less than 0.05 was considered statistically significant.

## 3. Results

### 3.1. Torque

The insertion torque and intrabony length plots over time revealed that the insertion torque rose slowly as the tip of the miniscrew was inserted into the cortical bone at the beginning of the implantation (Figure 4). The insertion torque rose more quickly as the straight middle portion of the miniscrew started to enter the bone. Once the tip portion had penetrated the cortical bone completely, and the contact area between the bone and miniscrews remained constant, the insertion torque increased slightly in low-density cortical bone and decreased slightly in the high-density samples. Furthermore, the insertion time increased with increasing cortical bone density.

The MIT value of different intrabony lengths (3, 4, 5, and 6 mm) was 5.37 ± 0.39, 6.33 ± 0.54, 6.97 ± 0.50, and 7.47 ± 0.46 Ncm for 30 pcf groups; 8.37 ± 0.54, 9.62 ± 0.56, 9.27 ± 0.39, and 9.67 ± 0.59 Ncm for 40 pcf groups; 13.05 ± 0.45, 13.85 ± 0.50, 14.03 ± 0.60, and 14.13 ± 0.58 Ncm for 50 pcf groups, respectively (Figure 5A). The MRT value of different intrabony lengths (3, 4, 5, and 6 mm) was 4.00 ± 0.26, 5.87 ± 0.70, 6.63 ± 0.29, and 7.52 ± 0.39 Ncm for 30 pcf groups; 6.41 ± 0.50, 8.77 ± 0.27, 9.21 ± 0.42, and 10.12 ± 0.39 Ncm for 40 pcf groups; 11.30 ± 0.65, 13.80 ± 0.35, 14.15 ± 0.56, and 14.85 ± 0.42 Ncm for 50 pcf groups, respectively (Figure 5B). The MIT and MRT values of the miniscrews at the various intrabony lengths were higher in the denser cortical bone, thereby demonstrating that the torque was readily affected by the cortical bone density (*p* < 0.05). The MIT and MRT values of the miniscrews in 30 pcf cortical bone increased with an increasing intrabony length. A similar trend was observed in the MRTs of the 40 and 50 pcf groups, while there was no significant difference between the MIT values at 4, 5, and 6 mm (*p* < 0.05).

### 3.2. Removal Angular Momentum

The removal torque plot over time was smoother during removal from high-density cortical bone, indicating that the removal torque decreased at a slower rate compared to the low-density samples (Figure 6). As the intrabony length was increased at a specific cortical bone density, the removal torque curve decreased slowly. Furthermore, the removal torque dropped rapidly at an intrabony length of 3 mm.

The RAM over the first rotation during removal (0–12 s) was calculated by integrating the torque over time (Figure 7A). The RAM value of different intrabony lengths (3, 4, 5, and 6 mm) was 19.19 ± 2.17, 45.45 ± 2.76, 63.02 ± 3.89, and 75.19 ± 2.84 Ncm·s for 30 pcf groups; 29.22 ± 2.28, 70.45 ± 4.85, 91.36 ± 4.00, and 103.42 ± 3.29 Ncm·s for 40 pcf groups; 59.67 ± 2.95, 124.13 ± 12.83, 144.64 ± 3.18, and 153.67 ± 2.73 Ncm·s for 50 pcf groups, respectively (Figure 7B). The energy required to remove a miniscrew from the high-density cortical bone was significantly higher than from the low-density samples (*p* < 0.05). The RAM at different intrabony lengths also varied significantly among the samples at a specific density (*p* < 0.05). 

### 3.3. Horizontal Resistance

The horizontal force was plotted against the deflection distance (Figure 8A), where the horizontal forces at a deflection of 0.6 mm were compared (Figure 8B). The horizontal force value of different intrabony lengths (3, 4, 5, and 6 mm) was 37.80 ± 4.36, 42.11 ± 2.21, 64.68 ± 5.65, and 79.18 ± 5.47 N for 30 pcf groups; 47.63 ± 7.58, 70.86 ± 4.43, 101.28 ± 16.59, and 94.21 ± 9.36 N for 40 pcf groups; 77.24 ± 15.45, 92.61 ± 10.44, 131.02 ± 9.77, and 130.50 ± 13.87 N for 50 pcf groups, respectively (Figure 8B). The miniscrews in high-density cortical bone exhibited a higher horizontal resistance compared to those in low-density bone (*p* < 0.05). Furthermore, the horizontal resistance increased with an increasing intrabony length at a lower density of 30 pcf, while no significant difference was observed between the miniscrews at an intrabony length of 5 and 6 mm in the 40 and 50 pcf cortical bone (*p* < 0.05).

### 3.4. Micromotion

PTV and IST are commonly used to evaluate micromotion, where a previous report [22] has stated that a lower PTV is indicative of less micromotion and is thus a marker of higher primary stability. Furthermore, a higher IST value is indicative of higher primary stability. The PTV value of different intrabony lengths (3, 4, 5, and 6 mm) was 11.41 ± 1.59, 9.22 ± 1.52, 7.58 ± 1.17, and 7.09 ± 0.83 for 30 pcf groups; 9.26 ± 0.92, 6.94 ± 0.47, 6.31 ± 0.87, and 6.44 ± 1.25 for 40 pcf groups; 5.64 ± 1.04, 5.15 ± 0.66, 4.66 ± 0.29, and 4.74 ± 1.31 for 50 pcf groups, respectively (Figure 9A). The IST value of different intrabony lengths (3, 4, 5, and 6 mm) was 49.92 ± 0.58, 51.42 ± 1.11, 52.17 ± 1.89, and 52.08 ± 0.66 for 30 pcf groups; 51.67 ± 1.17, 53.92 ± 1.11, 54.00 ± 0.55, and 53.67 ± 1.57 for 40 pcf groups; 55.00 ± 1.48, 55.00 ± 0.63, 56.17 ± 0.98, and 56.08 ± 1.32 for 50 pcf groups, respectively (Figure 9B). The PTV and IST values revealed that micromotion in the high-density cortical bone was lower than that in low-density samples (*p* < 0.05). The miniscrews with a longer intrabony length exhibited less micromotion in the low-density cortical bone, but there was no significant difference between the miniscrews at different intrabony lengths in the high-density cortical bone (50 pcf) (*p* < 0.05).

## 4. Discussion

The widespread use of miniscrews in orthodontic treatment has been accompanied by a continued improvement in success rate and stability, as innovative materials and designs are introduced [5,23]. However, miniscrews are still associated with a failure rate of 13.5 to 14% [4]. Adequate primary stability is indicative of stronger miniscrew–bone integration, which is beneficial for tissue regeneration to form substantial secondary stability [4]. The primary stability of a miniscrew is related to its geometric design (i.e., diameter, length, and thread form), as well as bone quality, periodontal tissue inflammation, and operation skills [24]. However, the effect of the miniscrew length on primary stability remains unknown. Previous studies have reported that the bone quality, implant site preparation, and miniscrew diameter have a more significant effect on primary stability than miniscrew length [25,26,27,28]. In clinical treatment, longer miniscrews are associated with a higher risk of proximity to the roots, damage to adjacent tissues, and even bicortical sinus perforations [29]. Thus, shorter miniscrews are advantageous, especially when implanted in the narrow space of the interseptum of the alveolar bone (e.g., posterior molar region of the mandible) [8]. However, most previous reports state that the miniscrew length had no significant effect on the success rate at lengths longer than 6 mm. Suzuki et al. [8] reported that miniscrews with a length of 5 mm were associated with a higher failure rate than longer miniscrews. As most previous studies focused on clinical studies with many other variables, this study evaluated miniscrews in in vitro experiments under variable controllable conditions to provide more information on this issue.

One of the limitations of this study was the difficulty in the commercial purchase of miniscrews with a length of less than 5 mm. However, to minimize the deviation caused by the lever effect while measuring the micromotion and horizontal resistance, both the tapping and pressing positions were selected to be close to the upper part of the bone–miniscrew interface.

Human alveolar bone comprises denser cortical bone and a relatively soft and porous cancellous bone. Bone quality and porosity differ among patients, while the cortical bone density varies along the vertical skeletal facial profile and in different locations [30]. In the present study, a range of implant environments was simulated by producing artificial bone samples to represent different cortical bone densities. The use of artificial bone can minimize the bias associated with uneven density and thickness and avoids the gradual change in the bone quality of human cadavers and fresh animal bones [31]. Unfortunately, there is no recognized benchmark for selecting substitute materials for alveolar bone for in vitro study. Although polyurethane foam blocks are similar to human bone in terms of density and stress–strain behavior, their porous structure and lower mechanical properties result in a lower thrust and torque during drilling compared with human or animal bone [32,33,34].

The MIT values were generally affected by both the intrabony length and cortical bone density (Figure 5A). However, the intrabony length had less of an effect in more dense cortical bone, where no significant differences were observed between the 4, 5, and 6 mm samples in the 40 and 50 pcf bone. Once the tip of the miniscrew had completely penetrated the cortical bone, there was a slight increase in the MIT values during insertion into 30 pcf cortical bone density, while the values remained relatively constant in the higher density bone (Figure 4). Although most previous studies have reported that cortical bone is a key factor affecting primary stability, Marquezan et al. [35] reported that cancellous bone also has an effect. The insertion torque during implantation is affected by several factors attributed to either the cortical bone or the cancellous bone. The contact area between the cancellous bone and miniscrew increased with increasing intrabony length, resulting in an increase in torque. However, wear of cortical bone during implantation at a specific contact area between the cortical bone and miniscrews leads to a decrease in insertion torque. The cancellous bone had a more significant impact on MIT in the lower density bone samples, indicating an increase in insertion torque. However, the denser cortical bone exhibited a decrease in MIT value that was equivalent to the increase observed in the cancellous bone, resulting in a stable insertion torque. 

Micromotion has become one of the most trusted methods for evaluating primary stability in recent years [12,17,36]. Miniscrew micromotion profoundly affects bone regeneration [17], where a small degree of micromotion is vital for active bone reconstruction. Micromotion measuring devices, such as PTV and IST, are portable and easy to operate, allowing simultaneous measurement of micromotion during treatment, leading to their rapid promotion in the field of oral treatment. The evaluation of the micromotion revealed a similar trend to the insertion torque (Figure 9), where the intrabony length had a more significant effect on the low-density cortical bone samples than those with a higher density. A large degree of micromotion can destroy the tissue between the bone and miniscrew, as osteoclasts tend to occupy the space, which can interfere with bone reconstruction and compromise the stability [37]. Hand-held micromotion measurement devices such as PTV and IST are portable and easy to use and have become popular tools for the evaluation of miniscrew micromotion. Previous research has indicated that micromotion is affected by the cortical bone density and porosity, which supports the findings of the current study [18]. However, the accuracy of these non-invasive easy-to-operate methods for evaluating primary stability is still debated [17,31].

Horizontal resistance is known to affect primary stability, where a miniscrew displacement larger than 0.6 mm is indicative of failure risk in clinical treatment [38]. Some studies have demonstrated that the application of a continuous horizontal load to a miniscrew leads to a maximum stress concentrated around the neck of the bone–miniscrew interface [39,40]. The difference in the density of cortical and cancellous bone allows for horizontal traction to produce a moment with the bottom of the cortical bone as the fulcrum. This leads to a tilting of the miniscrew. Therefore, the horizontal resistance is affected by the cortical bone, cancellous bone, and intrabony length. Extension of the force arm on the load side leads to a lower horizontal resistance, where the displacement of the miniscrew is dependent on the elastic deformation of the miniscrew itself and changes in the neck area of the cortical bone when the arm on the cancellous bone side is longer [18].

The MRT results exhibited a different trend, where the intrabony length affected all samples, including the high-density cortical bone group (11.30 ± 0.65, 13.80 ± 0.35, 14.15 ± 0.56, and 14.85 ± 0.42 Ncm for 3, 4, 5, and 6 mm intrabony lengths, respectively, *p* < 0.05) (Figure 5B). Thus, the cancellous bone led to resistance during removal, which was confirmed by the resulting removal torque curve and RAM results (Figure 7B). Regardless of the cortical bone density, the rate at which the removal torque decreased dropped significantly as the intrabony length increased. Thus, RAM was found to be more significantly affected by the intrabony length than MIT and MRT (Figure 7B). More energy was consumed during the first rotation of the miniscrew removal when the intrabony length was longer, which was accompanied by a higher remaining torque compared to the miniscrews with a shorter intrabony length. A continuous counterclockwise orthodontic force and moment is applied to loosen the miniscrews during clinical treatment [41]. Therefore, the success of miniscrew implantation relies on providing high resistance to maintain a higher torque. MIT and MRT represent instantaneous states, while RAM is a more reliable indicator of success, as it represents the energy consumed over a period of time. This also explains why shorter miniscrews in clinical treatments exhibit similar values to longer miniscrews during stability testing (e.g., MIT and micromotion) but are associated with a higher failure rate [8].

The 3 mm intrabony length miniscrews exhibited a significantly lower stability than the other groups throughout testing. The tip of the miniscrews did not completely penetrate the cortical bone when the implantation depth was only 3 mm, which led to a significantly lower bone–implant area compared to the other groups. This affected the stability and success rate of the miniscrews. The miniscrews with intrabony lengths of 5 and 6 mm exhibited similar performance in most tests, with the exception of RAM. This demonstrated the application potential of slightly shorter miniscrews (5 mm) in cases with lower traction, lower muscle activity, and higher cortical bone density. Furthermore, the use of even shorter miniscrews (3 mm) could be realized by shortening the length of the tip while maintaining the penetration ability.

Artificial bone was used in this study to control the variables and ensure the consistency of the results. However, the physical integrity and chemical composition were quite different from those of human bones. Furthermore, artificial bone cannot simulate the biological response of bones to torque and thermal changes. Moreover, with in vitro experiments, the implantation method, rotation speed, and environmental temperature are relatively different from the conditions involved in clinical treatment. Therefore, the significance of these results is to provide essential data for clinical research, and our findings should not be regarded as a benchmark. It is recommended that these findings be validated in further ex vivo or in vivo studies.

## 5. Conclusions

This study demonstrated that the primary stability of miniscrews was affected by both the cortical bone density and intrabony length, where the influence of the intrabony length was more pronounced at lower cortical bone densities. Regardless of the cortical bone density, more energy was required to remove the miniscrews as the implantation depth increased, indicating higher resistance and less risk of falling out. However, the artificial bone used in this study provided a limited simulation of the complex bone environment, and further research is recommended.

## Figures and Tables

**Figure 1 materials-13-05615-f001:**
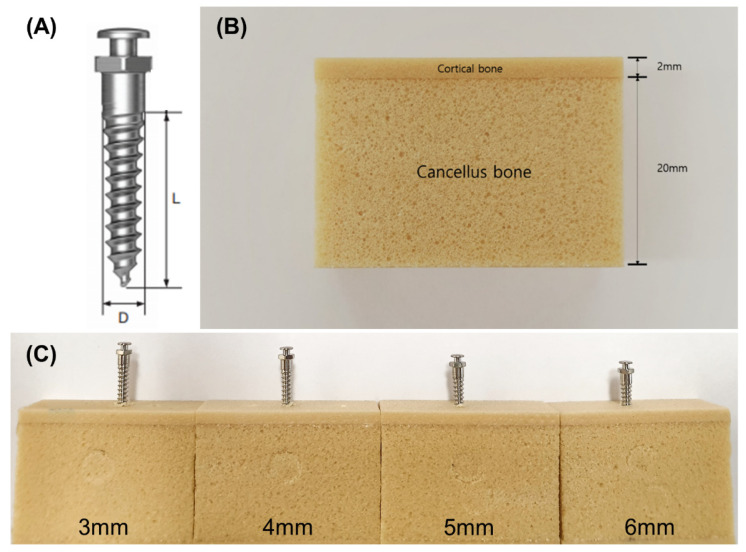
(**A**) Schematic diagram of the miniscrew, where diameter (D) = 1.8 mm and length (L) = 10 mm. Digital photographs of (**B**) an artificial bone block, and (**C**) miniscrews inserted at different intrabony lengths.

**Figure 2 materials-13-05615-f002:**
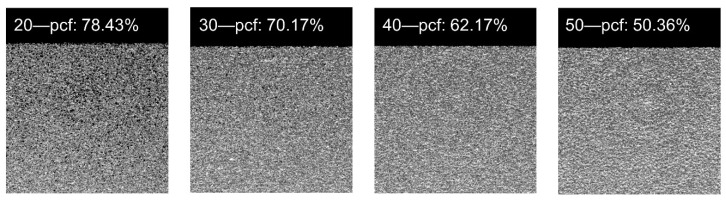
Porosity of the polyurethane foam blocks with different densities.

**Figure 3 materials-13-05615-f003:**
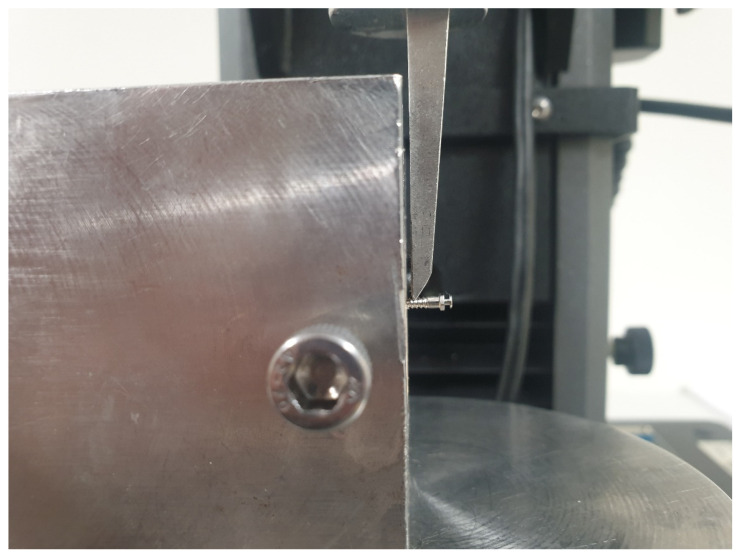
Digital photograph of the horizontal resistance test set-up.

**Figure 4 materials-13-05615-f004:**
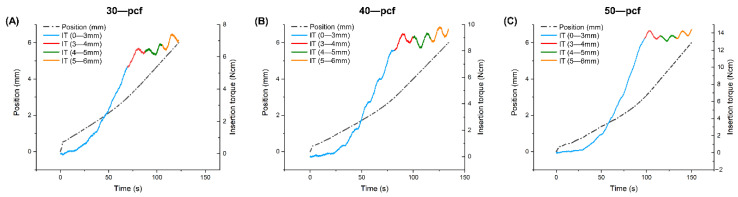
Position and insertion torque (IT) during insertion of the miniscrews into (**A**) 30, (**B**) 40, and (**C**) 50 pcf artificial cortical bone.

**Figure 5 materials-13-05615-f005:**
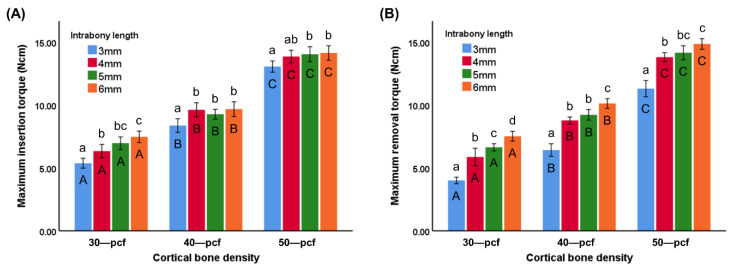
(**A**) Maximum insertion torque (MIT) and (**B**) maximum removal torque (MRT) of the miniscrews at various intrabony lengths in cortical bone of different densities. Different uppercase letters indicate a significant difference between the miniscrews inserted into cortical bone of various densities (*p* < 0.05), while different lowercase letters indicate a significant difference between the miniscrews at the various intrabony lengths (*p* < 0.05).

**Figure 6 materials-13-05615-f006:**
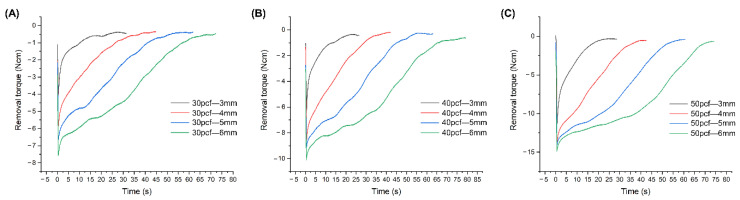
Removal torque values of time in (**A**) 30, (**B**) 40, and (**C**) 50 pcf cortical bone.

**Figure 7 materials-13-05615-f007:**
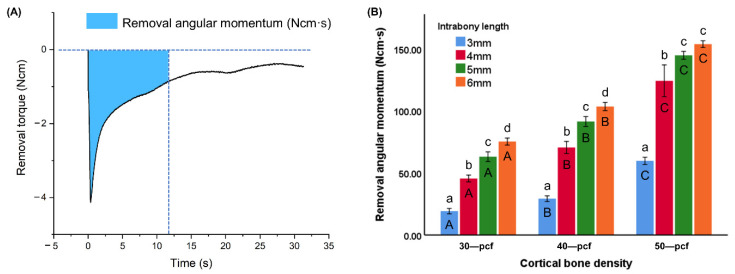
(**A**) Removal torque over time with removal angular momentum (blue). (**B**) Removal angular momentum (RAM) of miniscrews at different intrabony lengths in varying cortical bone densities. Different uppercase letters indicate a significant difference between the miniscrews inserted into cortical bone of various densities (*p* < 0.05), while different lowercase letters indicate a significant difference between the miniscrews at the various intrabony lengths (*p* < 0.05).

**Figure 8 materials-13-05615-f008:**
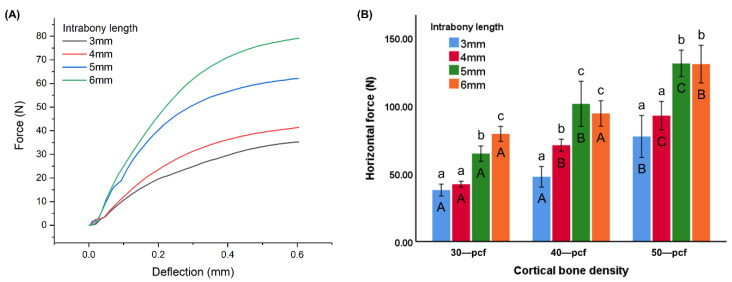
Horizontal resistance represented as (**A**) horizontal force versus deflection distance and (**B**) horizontal force values at 0.6 mm deflection. Different uppercase letters indicate a significant difference between the miniscrews inserted into cortical bone of various densities (*p* < 0.05), while different lowercase letters indicate a significant difference between the miniscrews at the various intrabony lengths (*p* < 0.05).

**Figure 9 materials-13-05615-f009:**
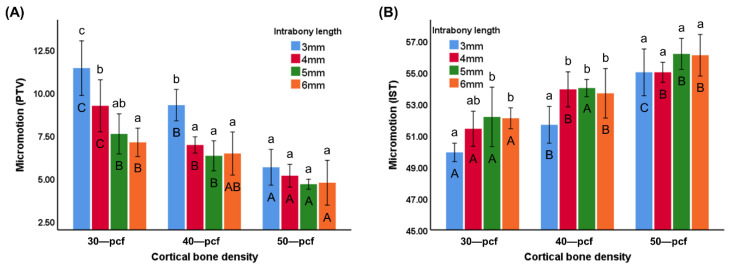
Micromotion represented as (**A**) periotest value (PTV) (**B**) implant stability testing value (IST). Different uppercase letters indicate a significant difference between the miniscrews inserted into cortical bone of various densities (*p* < 0.05), while different lowercase letters indicate a significant difference between the miniscrews at the various intrabony lengths (*p* < 0.05).

**Table 1 materials-13-05615-t001:** Mechanical properties of the polyurethane foam bone blocks with different densities.

Density		Compression		Tension	
pcf	(g/cc)	Strength (MPa)	Modulus (MPa)	Strength (MPa)	Modulus (MPa)
**20**	0.32	8.4	210	5.6	284
**30**	0.48	18	445	12	592
**40**	0.64	31	759	19	1000
**50**	0.80	48	1148	27	1469

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
