# Peer review of "Effects of Intrabony Length and Cortical Bone Density on the Primary Stability of Orthodontic Miniscrews"

_materials, 2020, doi:10.3390/ma13245615_

Round 1

Reviewer 1 Report

Jin et al. have prepared a very well-documented, technical paper studying the effect of miniscrew length on primary stability. The authors evaluated several physical properties using foam blocks that mimic bones with various densities. Also, they used 10 mm screws with various lengths of insertion to study their stability. The paper is well written and follows a systematic approach. However, I feel that it might benefit from some improvements:

  • the impact of the miniscrew free part length should be addressed, as the lever effect is involved; this can influence both the results of the stability  of the intrabony component, but also the overall retention of the screw; in this regard, a discussion whether shorter screws might have different stabilities than longer screws that are not inserted completely should be made
  • is the drill/insertion method a key factor to be considered in the testing?
  • how was the threadform chosen for this particular miniscrew that was used in the testing? this might play a more important role than length, and might be a confounding variable in the study
  • some language errors throughout the manuscript should be corrected

Respectfully submitted.

Author Response

General comment

Jin et al. have prepared a very well-documented, technical paper studying the effect of miniscrew length on primary stability. The authors evaluated several physical properties using foam blocks that mimic bones with various densities. Also, they used 10 mm screws with various lengths of insertion to study their stability. The paper is well written and follows a systematic approach. However, I feel that it might benefit from some improvements:

Q1

The impact of the miniscrew free part length should be addressed, as the lever effect is involved; this can influence both the results of the stability of the intrabony component, but also the overall retention of the screw; in this regard, a discussion whether shorter screws might have different stabilities than longer screws that are not inserted completely should be made.

A1

Thank you for the valuable suggestions. The relevant information has been added according to your comments as follows:

2.4 Horizontal resistance

In the current measurement, the loading position was selected to be the same position, close to the upper part of the bone–miniscrew interface, to minimize the deviation caused by the lever effect arising from the variation in the length of the free part.

2.5 Micromotion

The artificial bone was fixed using a metal clamp, and the devices were set at a fixed distance from the miniscrew (n = 6) surface, in a direction perpendicular to its long axis. In the current measurement, the tapping position was selected to be the same position, close to the upper part of the bone-miniscrew interface, to minimize the deviation caused by the lever effect arising from the variation in the length of the free part.

4. Discussion

One of the limitations of this study was the difficulty in the commercial purchase of miniscrews with a length of less than 5 mm. However, to minimize the deviation caused by the lever effect while measuring the micromotion and horizontal resistance, both the tapping and pressing positions were selected to be close to the upper part of the bone-miniscrew interface.

Q2

Is the drill/insertion method a key factor to be considered in the testing?

A2

The implantation and removal procedure are the method according to ASTM F-1839-08 and is commonly used in previous in vitro studies. The relevant information has been added according to your comments as follows:

2.2 Torque

Torque testing was conducted according to a previously reported procedure [18,20]. The artificial bone was fixed with a metal clamp, and the miniscrews (n = 6) were implanted in a clockwise manner using an automatic torque device (Admet eXpert8600, ADMET, Norwood, WA, USA) with a self-drilling system at 5 rpm, according to ASTM F-1839-08. The implantation depth (3 cm, 4 cm, 5 cm, and 6 cm) was controlled by a supporting software (GaugeSafe software, ADMET, Norwood, WA, USA). Removal was performed in a counterclockwise manner at 5 rpm. The torque was measured every second during implantation and removal using the supporting software, and the maximum insertion torque (MIT) and maximum removal torque (MRT) were recorded.

Q3

How was the threadform chosen for this particular miniscrew that was used in the testing? this might play a more important role than length, and might be a confounding variable in the study

A3

According to your suggestion, we have added the relevant information about the miniscrew in the materials section as follows:

2.1 Materials

Cylindrical-type orthodontic titanium miniscrews were used with no surface treatment (machined surface, OSSH1810, Osstem Implant, Busan, Korea) and a diameter, pitch, length, and head height of 1.8 mm, 0.7 mm, 10 mm, and 1.95 mm, respectively (Figure 1A).

Reviewer 2 Report

This experimental study was designed to investigate the mechanical characteristics of miniscrew implantation for orthodontic purposed by a bone simulator model.

General comment

1) In my opinion, use of a purportedly bone-like plastic foam is a major flaw of this otherwise accurately performed experiment: in fact, evidence that this artificial material actually behaves as true bone is limited. Therefore, the reported findings appear poorly related to the clinical situation and of limited value to address the issues raised by miniscrew placement for orthodontic purposes. In this context, a crucial point for successful placement of miniscrews is related to their osseoconductive surface, which can deeply influence their interaction with bone cells and hence intra-bony stabilization. Obviously, this aspect of the matter cannot be elucidated by the chosen artificial model.

Specific comments

2) In the introduction, information on polyurethane foam blocks as suitable bone substitute for experimental purposes is missing, as is the relevant literature needed to validating this model. In the Methods section, the mechanical properties of both materials, namely polyurethane foam and bone, must be reported in detail in order to allow comparison (e.g. data relevant to authentic bone should be included in table 1).

3) Is the procedure (rpm, torque) used to screw/unscrew the miniscrews similar to that employed in orthodontics?

4) Additional details of the micromotion experiment should be provided to allow understanding by a general readership.

5) The English language is quite good, although revision by a mother-tongue specialist is advisable

Author Response

Materials

Special Issue - Biomaterials for Medical and Dental Application

Re: Revision of “Effects of Intrabony Length and Cortical Bone Density on the Primary Stability of Orthodontic Miniscrews” (materials-1021325)

Thank you very much for your kind reviews and comments regarding our manuscript (materials-1021325) entitled above. We have conducted revisions according to your comments, and we hope this will be adequate for the acceptance of this manuscript. The revised text is highlighted in red. Details of corrections according to the comments are as follows, and the English language of this article was corrected.

Response to Reviewer #2

General comment

This experimental study was designed to investigate the mechanical characteristics of miniscrew implantation for orthodontic purposed by a bone simulator model. In my opinion, use of a purportedly bone-like plastic foam is a major flaw of this otherwise accurately performed experiment: in fact, evidence that this artificial material actually behaves as true bone is limited. Therefore, the reported findings appear poorly related to the clinical situation and of limited value to address the issues raised by miniscrew placement for orthodontic purposes. In this context, a crucial point for successful placement of miniscrews is related to their osseoconductive surface, which can deeply influence their interaction with bone cells and hence intra-bony stabilization. Obviously, this aspect of the matter cannot be elucidated by the chosen artificial model.

Q1

In the introduction, information on polyurethane foam blocks as suitable bone substitute for experimental purposes is missing, as is the relevant literature needed to validating this model. In the Methods section, the mechanical properties of both materials, namely polyurethane foam and bone, must be reported in detail in order to allow comparison (e.g. data relevant to authentic bone should be included in table 1).

A1

Thank you for your thoughtful comments on the manuscript. Your valuable opinions have assisted me to gain new knowledge related to this field and to have a clearer understanding of the limitations in this research. According to your suggestion, we have added sentences related to polyurethane foam blocks in the introduction, methods, and discussion sections.

1. Introduction

However, most previous studies have employed miniscrews larger than 6 mm. Therefore, an in vitro study is necessary to evaluate the effect of the length of miniscrews when the insertion depth is less than 6 mm, which can then provide a basis for subsequent clinical studies. Polyurethane foam is one of the substrates often used to simulate human cortical bone and cancellous bone [12-14]. In biomechanical experiments, polyurethane foam can maintain homogeneity and consistency, which assists with the elimination of bias caused by other factors, and exhibits stress-strain behavior similar to that of human bone, making it suitable for preclinical evaluation of miniscrews [12].

2.1 Materials

According to standard procedure (ASTM F-1839-08), polyurethane foam blocks (Sawbone, Division of Pacific Research Laboratories, Inc., Vashon, WA USA) with densities of 20, 30, 40, and 50 pounds per cubic foot (pcf)—equivalent to densities ranging from 0.32–0.80 g/cc—were used to represent cancellous bone and cortical bone (Table 1). This range of densities correlates with the human maxillary cortical bone density (ranging from 0.31–1.11 g/cc) reported by Devlin et al. [15] and the human mandibular bone density (average 0.664 g/cc) reported by Kido et al. [16].

In addition, based on your valuable comments, we are trying to compare the mechanical properties of polyurethane foam and bone, but the compressive testing of polyurethane foam was conducted according to the ASTM standard F1839-08, which requires large specimens (50.8 mm × 50.8 mm × 25.4 mm blocks), and no specimens of this specification can be found in human bones (particularly for cortical bone). Therefore, the size of the test piece is different, and as such the data cannot be directly compared. However, according to the literature, the porous structure and lower mechanical properties of polyurethane foam result in a lower thrust and torque when drilling compared to human or animal bone. In the manuscript, we have supplemented this limitation.

4. Discussion

Unfortunately, there is no recognized benchmark for selecting substitute materials for alveolar bone for in vitro study. Although polyurethane foam blocks are similar to human bone in terms of density and stress-strain behavior, their porous structure and lower mechanical properties result in a lower thrust and torque during drilling compared with human or animal bone [32-34].

Moreover, with in vitro experiments, the implantation method, rotation speed, and environmental temperature are relatively different from the conditions involved in clinical treatment. Therefore, the significance of these results is to provide essential data for clinical research, and our findings should not be regarded as a benchmark. It is recommended that these findings be validated in further ex vivo or in vivo studies.

Q2

Is the procedure (rpm, torque) used to screw/unscrew the miniscrews similar to that employed in orthodontics?

A2

The miniscrews were inserted using an automatic torque device with a self-drilling system at a speed of 5 rpm according to ASTM F-1839-08, which states that 3–5 rpm is appropriate. However, according to the literature reviewed, this rotation speed is slower than the rotation speed in clinical treatment (15 rpm) [3]. In the follow-up study, we hope to also evaluate the influence of rotation speed on stability. Some related sentences were modified as follows:

4. Discussion

Moreover, with in vitro experiments, the implantation method, rotation speed, and environmental temperature are relatively different from the conditions involved in clinical treatment. Therefore, the significance of these results is to provide essential data for clinical research, and our findings should not be regarded as a benchmark. It is recommended that these findings be validated in further ex vivo or in vivo studies.

Q3

Additional details of the micromotion experiment should be provided to allow understanding by a general readership.

A3

According to your suggestion, the related sentences were added in the methods and discussion sections.

2.5 Micromotion

Micromotion analysis was conducted according to a previously reported procedure [18]. The periotest value (PTV) tester (Medizintechnik Gulden, Modautal, Germany) and implant stability tester (IST) (AnyCheck, Neo Biotech, Seoul, Korea) are two commercial hand-held micromotion measurement devices that obtain micromotion data by repeatedly tapping the head of a miniscrew. The artificial bone was fixed with a metal clamp, and the devices were set at a fixed distance from the miniscrew (n = 6) surface, in a direction perpendicular to its long axis. In the current measurement, the tapping position was selected to be the same position close to the upper part of the bone-miniscrew interface to minimize the deviation caused by the lever effect arising from the variation in the length of the free part. The devices were calibrated before each measurement according to the manufacturer’s instructions  [22]. Measurements were performed in triplicate, and the mean values have been reported.

4. Discussion

Micromotion has become one of the most trusted methods for evaluating primary stability in recent years [12,36,37]. Miniscrew micromotion profoundly affects bone regeneration [17], where a small degree of micromotion is vital for active bone reconstruction. Micromotion measuring devices, such as PTV and IST, are portable and easy to operate, allowing simultaneous measurement of micromotion during treatment, leading to their rapid promotion in the field of oral treatment.

Reviewer 3 Report

The authors should at least mention, for each experimental test, the number of implanted miniscrews (1 ? 10 ? 100 ?..).

The statistical analysis of the results for each experimental test group should also be presented and discussed.

Author Response

Materials

Special Issue - Biomaterials for Medical and Dental Application

Re: Revision of “Effects of Intrabony Length and Cortical Bone Density on the Primary Stability of Orthodontic Miniscrews” (materials-1021325)

Thank you very much for your kind reviews and comments regarding our manuscript (materials-1021325) entitled above. We have conducted revisions according to your comments, and we hope this will be adequate for the acceptance of this manuscript. The revised text is highlighted in red. Details of corrections according to the comments are as follows, and the English language of this article was corrected.

Response to Reviewer #3

Q1

The authors should at least mention, for each experimental test, the number of implanted miniscrews (1 ? 10 ? 100 ?..).

A1

Thank you for your valuable advice and thoughtful comments on the manuscript. A total of 216 miniscrews were used in this experiment. In each experiment, each group used 6 miniscrews (n = 6). Related sentences have been added in the methods section,

The artificial bone was fixed with a metal clamp, and the miniscrews (n = 6) were implanted in a clockwise manner using an automatic torque device…

A universal testing machine (Instron 5942, Instron, Norwood, MA, USA) was used to measure the horizontal resistance, where the artificial bone was fixed with a metal clamp while a knife-like shear jig applied a tangential load to the miniscrew (n = 6) at a crosshead speed of 1 mm/min (Figure 3).

The artificial bone was fixed with a metal clamp, and the devices were set at a fixed distance from the miniscrew (n = 6) surface, in a direction perpendicular to its long axis.

Q2

The statistical analysis of the results for each experimental test group should also be presented and discussed.

A2

According to your suggestion, statistical analysis of the results was added in the results and discussion section.

3. Results

The MIT value of different intrabony lengths (3, 4, 5, and 6 mm) was 5.37 ± 0.39, 6.33 ± 0.54, 6.97 ± 0.50, and 7.47 ± 0.46 Ncm for 30 pcf groups; 8.37 ± 0.54, 9.62 ± 0.56, 9.27 ± 0.39, and 9.67 ± 0.59 Ncm for 40 pcf groups; and 13.05 ± 0.45, 13.85 ± 0.50, 14.03 ± 0.60, and 14.13 ± 0.58 Ncm for 50 pcf groups, respectively (Figure 5A). The MRT value of different intrabony lengths (3, 4, 5, and 6 mm) was 4.00 ± 0.26, 5.87 ± 0.70, 6.63 ± 0.29, and 7.52 ± 0.39 Ncm for 30 pcf groups; 6.41 ± 0.50, 8.77 ± 0.27, 9.21 ± 0.42, and 10.12 ± 0.39 Ncm for 40 pcf groups; and 11.30 ± 0.65, 13.80 ± 0.35, 14.15 ± 0.56, and 14.85 ± 0.42 Ncm for 50 pcf groups, respectively (Figure 5B). The MIT and MRT values of the miniscrews at the various intrabony lengths were higher in the denser cortical bone, thereby demonstrating that the torque was readily affected by the cortical bone density (P < 0.05).

The RAM value of different intrabony lengths (3, 4, 5, and 6 mm) was 19.19 ± 2.17, 45.45 ± 2.76, 63.02 ± 3.89, and 75.19 ± 2.84 Ncm·s for 30 pcf groups; 29.22 ± 2.28, 70.45 ± 4.85, 91.36 ± 4.00, and 103.42 ± 3.29 Ncm·s for 40 pcf groups; and 59.67 ± 2.95, 124.13 ± 12.83, 144.64 ± 3.18, and 153.67 ± 2.73 Ncm·s for 50 pcf groups, respectively (Figure 7B). The energy required to remove a miniscrew from the high-density cortical bone was significantly higher than from the low-density samples (P < 0.05). The RAM at different intrabony lengths also varied significantly among the samples at a specific density (P < 0.05).

The horizontal force value of different intrabony lengths (3, 4, 5, and 6 mm) was 37.80 ± 4.36, 42.11 ± 2.21, 64.68 ± 5.65, and 79.18 ± 5.47 N for 30 pcf groups; 47.63 ± 7.58, 70.86 ± 4.43, 101.28 ± 16.59, and 94.21 ± 9.36 N for 40 pcf groups; and 77.24 ± 15.45, 92.61 ± 10.44, 131.02 ± 9.77, and 130.50 ± 13.87 N for 50 pcf groups, respectively (Figure 8B). The miniscrews in high-density cortical bone exhibited a higher horizontal resistance compared to those in low-density bone (P < 0.05). Furthermore, the horizontal resistance increased with an increasing intrabony length at a lower density of 30 pcf, while no significant difference was observed between the miniscrews at an intrabony length of 5 and 6 mm in the 40 and 50 pcf cortical bone (P < 0.05).

The PTV value of different intrabony lengths (3, 4, 5, and 6 mm) was 11.41 ± 1.59, 9.22 ± 1.52, 7.58 ± 1.17, and 7.09 ± 0.83 for 30 pcf groups; 9.26 ± 0.92, 6.94 ± 0.47, 6.31 ± 0.87, and 6.44 ± 1.25 for 40 pcf groups; and 5.64 ± 1.04, 5.15 ± 0.66, 4.66 ± 0.29, and 4.74 ± 1.31 for 50 pcf groups, respectively (Figure 9A). The IST value of different intrabony lengths (3, 4, 5, and 6 mm) was 49.92 ± 0.58, 51.42 ± 1.11, 52.17 ± 1.89, and 52.08 ± 0.66 for 30 pcf groups; 51.67 ± 1.17, 53.92 ± 1.11, 54.00 ± 0.55, and 53.67 ± 1.57 for 40 pcf groups; and 55.00 ± 1.48, 55.00 ± 0.63, 56.17 ± 0.98, and 56.08 ± 1.32 for 50 pcf groups, respectively (Figure 9B). The PTV and IST values revealed that micromotion in the high-density cortical bone was lower than that in low-density samples (P < 0.05). The miniscrews with a longer intrabony length exhibited less micromotion in the low-density cortical bone, but there was no significant difference between the miniscrews at different intrabony lengths in the high-density cortical bone (50 pcf) (P < 0.05).

4. Discussion

The MRT results exhibited a different trend, where the intrabony length affected all samples, including the high-density cortical bone group (11.30 ± 0.65, 13.80 ± 0.35, 14.15 ± 0.56, and 14.85 ± 0.42 Ncm for 3, 4, 5, and 6 mm intrabony lengths, respectively, P < 0.05).

Round 2

Reviewer 2 Report

The authors have done a commendable additional work on their manuscript and have satisfactorily replied to my criticisms to the previous version. They have properly underscored the actual value as well as the limitations of their results for orthodontic clinical practice.

Reviewer 3 Report

After this extensive correction and adaptation to the reviewers comments, the paper can now be edited